# Regulation of Ribosome Function by RNA Modifications in Hematopoietic Development and Leukemia: It Is Not Only a Matter of m^6^A

**DOI:** 10.3390/ijms22094755

**Published:** 2021-04-30

**Authors:** Francesco Fazi, Alessandro Fatica

**Affiliations:** 1Department of Anatomical, Histological, Forensic & Orthopedic Sciences, Section of Histology & Medical Embryology, Sapienza University of Rome, 00165 Rome, Italy; 2Istituto Pasteur Italia-Fondazione Cenci Bolognetti, 00161 Rome, Italy; 3Department of Biology and Biotechnology ‘Charles Darwin’, Sapienza University of Rome, 00165 Rome, Italy

**Keywords:** epitranscriptomics, RNA modifications, tRNA, rRNA

## Abstract

Growth and maturation of hematopoietic stem cells (HSCs) are largely controlled at both transcriptional and post-transcriptional levels. In particular, hematopoietic development requires a tight control of protein synthesis. Furthermore, translational deregulation strongly contributes to hematopoietic malignancies. Researchers have recently identified a new layer of gene expression regulation that consists of chemical modification of RNA species, which led to the birth of the epitranscriptomics field. RNA modifications provide an additional level of control in hematopoietic development by acting as post-transcriptional regulators of lineage-specific genetic programs. Other reviews have already described the important role of the N^6^-methylation of adenosine (m^6^A) within mRNA species in regulating hematopoietic differentiation and diseases. The aim of this review is to summarize the current status of the role of RNA modifications in the regulation of ribosome function, beyond m^6^A. In particular, we discuss the importance of RNA modifications in tRNA and rRNA molecules. By balancing translational rate and fidelity, they play an important role in regulating normal and malignant hematopoietic development.

## 1. Introduction

The identification of chemically modified nucleotides within RNA molecules dates back to the early 1950s [1]. Recently, more than 160 different chemical RNA modifications have been discovered [2]. Initially, they were mostly identified in highly expressed RNA species, such as ribosomal RNAs (rRNAs) and transfer RNAs (tRNAs), with fully digested RNA followed by mass spectrometry. However, this bulk technique does not allow assignment of the modified nucleotide to specific RNA molecules. The recent development of novel methodologies based on high-throughput sequencing for mapping modifications within specific RNA sequences, including low-abundance molecules such as messenger RNAs (mRNAs), has produced a huge breakthrough in the field [3]. These sequencing approaches use antibodies to immunoprecipitate modified RNAs or specific chemical treatments to render the modified nucleotide visible by producing mutations or truncations in the preparation of cDNA. These methods have allowed, for the first time, mapping of unique modifications within the cell transcriptome. However, sequencing methodologies have also generated conflicting results, especially for the identification of chemical modifications within low-abundance RNA species, such as mRNA (reviewed in [4]). In conjunction with the epigenetic regulation of chromatin, the dynamic and reversible nature of some of these RNA modifications, and the fact that they can modulate gene expression without converting the sequence of the RNA molecules, brought about the creation of the epitranscriptomics field (also referred to as RNA epigenetics). However, an important difference with epigenetic modifications is that, at present, there is no evidence that they are heritable.

Most of the studies of the epitranscriptome in hematopoietic development and malignancies concern the N^6^-methyladenosine (m^6^A), which is the most abundant internal modification of mRNA. Several reviews have already described the important role of m^6^A-modified mRNAs in regulating genetic programs with relevant roles in normal hematopoiesis and leukemia [5,6,7,8,9,10,11]. However, within the epitranscriptome, modifications present in rRNAs and tRNAs largely prevail. Hematopoietic development requires tight control of protein synthesis [12,13]. Furthermore, translational deregulation strongly contributes to hematopoietic malignancies [14,15]. Therefore, by regulating ribosome function and translation fidelity, modification in tRNAs and rRNAs may have an equally important role. Here, we review the emerging function of RNA modifications, beyond m^6^A, occurring in tRNAs and rRNAs and their function in normal and malignant hematopoietic development (Table 1).

## 2. Balanced Ribosome Biogenesis and Translation Are Important for Hematopoietic Development

Hematopoiesis is the life-long process by which a small number of multipotent hematopoietic stem cells (HSCs) present in the bone marrow (BM) give rise to all the different hematopoietic cell types (review in [30]). This is a massive process (human BM produces about 10^12^ cells per day) that is regulated at multiple levels, such as epigenetic, transcriptional, post-transcriptional and translational control. In particular, ribosome biogenesis and translation are directly linked to the cell growth and differentiation capacity of HSCs [12]. Alteration in ribosome biogenesis and protein synthesis can affect both HSCs’ homeostasis and cell fate decisions. Single allele depletion of the ribosomal protein *Rpl24* produces a significant decrease in translation (about 30%) and impaired HSC differentiation [12]. However, the deletion of *Pten*, a regulator of the mTOR complex (see below) that produces an equivalent percentage increase in translation, also compromises HSC function [12]. These data indicate that both decrease and increase in translational level affect HSCs’ differentiation and that the proper translational rate is critical for hematopoietic development.

Ribosome biogenesis and translation are energy-demanding processes. For this reason, cells present different regulatory pathways that adapt ribosome levels and protein synthesis to growth conditions. One of these pathways is regulated by the mTOR complex I (mTORC1) (reviewed in [31]). This complex contains the mTOR1 kinase, which regulates both ribosome biogenesis and cap-dependent translation through phosphorylation and activation of ribosomal protein S6 kinase (*S6K*) and phosphorylation and inhibition of eIF4E-binding proteins (*4E-BPs*), respectively. mTORC1 is strictly required for differentiation of HSCs [32].

Furthermore, it was recently shown that differentiation of HSCs in myeloid progenitors requires not only a general increase in protein synthesis but also the translation of specific mRNAs [13]. Importantly, the alteration in ribosome levels that accompanies cell development can affect not only global protein synthesis but also the translation of specific transcripts. mRNAs with low translation initiation rates are particularly susceptible to a decrease in ribosome levels compared with mRNAs that are efficiently translated.

Altered translation or ribosome biogenesis can also promote hematopoietic malignancies. For example, the transcription factor *RUNX1*, which is frequently mutated in different types of human leukemia, is directly involved in the expression of ribosomal proteins (RPs), genes, and rRNA in HSCs [33]. Mutation in *Runx1* produces a reduction in ribosome levels, lower translation and stress-resistance. This confers a pre-malignancy state in which there is selective growth advantage and, eventually, accumulation of HSCs carrying *Runx1* mutation over normal HSCs [33]. However, in other cellular types, such as mesenchymal progenitor cells, the RUNX protein family was shown to inhibit rRNA transcription. In this case, the depletion of *Runx1* produced an increase in ribosome levels. This suggests that the effect of the RUNX1 protein is cell-type specific. Notably, in HSCs, the depletion of *Runx1* can be partially compensated by mTORC1 activation, indicating that ribosome biogenesis and translation provide an important contribution to the leukemogenic process [33].

In this context, it appears evident that an important role in hematopoietic development is played by chemical modifications of tRNA and rRNA molecules, which may impact both ribosome synthesis and translation.

## 3. Impact of RNA Modifications on Ribosome Function

### 3.1. tRNA Modifications

tRNA molecules are highly modified, and mutations in tRNA modification enzymes are responsible for several human diseases [34]. Modifications can occur along all the tRNA molecules and are important for translation accuracy and efficiency, and tRNA stability. In particular, modifications in the anticodon loop of tRNAs have long been known to affect codon recognition, while chemical modifications in the main body protect tRNA from stress-induced cleavages, which are responsible for the production of tRNA fragments (tRFs) [35]. tRFs regulate gene expression by changing mature tRNAs levels and by directly repressing translation through interaction with translation initiation factors and regulatory proteins, ribosomes, or mRNA structures [35].

HSCs are particularly sensitive to perturbations in protein synthesis levels (see above). Alteration in tRNA modifications may impact both HSCs quiescence and myeloid differentiation [36]. In HSCs, the RNA endonuclease angiogenin, which cleaves unmodified tRNAs, localizes in stress granules where it contributes to the maintenance of a quiescent cell state by producing tRFs that inhibit protein synthesis. During myeloid differentiation, angiogenin translocates in the nucleus where it promotes rRNA synthesis, increasing translation and proliferation levels [36]. As tRNA cleavage by angiogenin is tightly connected to tRNA modifications, it is conceivable that HSC quiescence might depend on specific tRNA modifications.

In vivo, the importance of tRNA modifications in hematopoiesis was demonstrated for DNMT2, the 5-methylcytosine (m^5^C) RNA methyltransferase responsible for modification of the C_38_ in the anti-codon loop of different tRNA species [16] (Figure 1). *Dnmt2* knockout mice exhibited defects in differentiation of bone marrow mesenchymal stromal cells (MSCs), which constitute an essential HSC niche component [16]. The absence of m^5^C_38_ does not change global protein translation but affects the translation of a specific mRNAs subset and produces codon mistranslation, indicating an important role for m^5^C in the regulation of codon fidelity. In particular, the mRNA encoding for nestin and periostin proteins was found to be negatively regulated at the translational levels by *Dnmt2* depletion. Notably, Nestin+ MSCs are tightly associated with HSCs in the bone marrow niche, and nestin-deficient mice showed significant reduction in HSCs. Moreover, the depletion of periostin in mice produced defects in the bones similar to those observed in *Dnmt2* knockout mice [37].

DNMT2 also regulates the production of tRNA fragments, which is also connected to mature tRNA levels (see above). Nevertheless, *Dnmt2* knockout mice did not present altered levels of tRNA molecules, despite the increase in tRNA fragments. It was suggested that this is due to the presence of a second m^5^C RNA methyltransferase, *NSun2* (NOP2/Sun RNA Methyltransferase 2), which maintains m^5^C levels in tRNA molecules. and *NSun2* double-knockout mice showed decreased tRNA levels and reduced translation [38]. However, any defects in hematopoiesis were not analyzed in the *Dnmt2-* and *NSun2*-deficient mouse. Notably, *NSun2* expression is positively regulated by the oncogene MYC [39], which plays important roles in hematopoiesis and is frequently upregulated in different types of leukemia [40].

Another link between m^5^C and hematopoiesis concerns the ten-eleven translocation 2 (*TET2*), a methylcytosine dioxygenase that converts m^5^C to 5-hydroxymethylcytosine (hm^5^C) in DNA. TET2 protein has been recently shown to be involved in the oxidation of m^5^C in both tRNA and mRNA species [17,18]. TET2 decreased m^5^C levels in tRNA molecules and increased the production of tRFs [41]. As such, it should counteract the activity of DNMT2 and NSUN2 on hematopoietic development. Nevertheless, it was shown that the activity of TET2 on mRNA promotes the development of myeloid cells, such as neutrophils and monocytes, during pathogen-mediated infection [17]. It was suggested that the depletion of TET2 increases the levels in mRNA by affecting double-stranded RNA formation, and thus the binding of regulatory proteins. In particular, it was shown that the m^5^C in the 3′-UTR of *Socs3* mRNA, a regulator of cytokine-induced myelopoiesis, inhibited the binding and, in turn, the destabilization promoted by ADAR1 [17]. However, TET2 only converts m^5^C to hm^5^C, and the enzyme responsible for the complete conversion of hm^5^C to cytosine in mRNA is still not known. Notably, TET2 is required for proper hematopoietic development in mice [18]. Moreover, the *TET2* gene is frequently mutated in myeloid malignancies, and the depletion of *Tet2* in mice led to myeloid transformation [18]. However, its role as a regulator of hematopoiesis and a tumor-suppressor gene has been mainly attributed to its DNA demethylating activity, and it is possible that some of the observed effects are indirectly due to the activity of TET2 on DNA. In conclusion, the impact of m^5^C on hematopoiesis and leukemogenesis warrants further investigation.

Another tRNA modification with a role in the maturation of blood cells is pseudouridylation (Ψ), which is one of the most abundant modifications in non-coding RNA molecules and is catalyzed by pseudouridine synthases (PUSs). The human genome encodes for 13 different PUSs, which directly recognize specific RNA sequences or structures, the only exception being dyskerin (*DKC1*, also known as *CBF5*), which requires antisense box H/ACA small nucleolar RNAs (snoRNAs) for substrate recognition (see below) [42]. Knockdown of the tRNA pseudouridine synthase *PUS7* in human CD34^+^ HSCs blocks hematopoietic differentiation in vitro, and the downregulation of *PUS7* in human HSCs strongly impaired their engraftment in recipient mice [21]. Notably, PUS7 levels were also found reduced in myelodysplastic syndromes (MDS), hematological disorders that may progress to acute myeloid leukemia (AML). Mechanistically, PUS7-mediated Ψ is required for the association between modified tRFs derived from tRNA containing a 5′ terminal oligoguanine (TOG) and the polyA binding protein (*PABPC1*), which results in repression of translation initiation [21]. Conversely, the presence of Ψ inhibits the association of tRFs with YBX1 and DHX36 proteins. Thus, it has been suggested that Ψ regulates conformational changes required for the association of tRFs with specific proteins. However, in these latter cases, the PUS7-mediated Ψ should have had a positive effect on protein synthesis because YBX1 and DHX36 (also known as RHAU) were found to stimulate mRNA translation in various studies [43,44]. This suggests that PUS7-mediated Ψ of tRNA might have the opposite effect on protein translation in different cellular contexts.

### 3.2. rRNA Modifications

Small (40S) and large (60S) ribosomal subunits contain one rRNA (18S) and three rRNAs (28S, 5.8S, and 5S), respectively. The 28S, 18S, and 5.8S rRNAs are transcribed by a single pre-rRNA precursor that is processed, modified, and assembled with ribosomal protein co-transcriptionally [45]. Ribosome maturation is a complex process that requires over 200 protein factors and hundreds of small nucleolar RNAs (snoRNA), which are part of small nucleolar ribonucleoprotein complexes (snoRNPs) [46]. snoRNPs are involved in both pre-rRNA processing and rRNA modifications [47]. The most abundant modifications of ribosomal RNAs (rRNAs) are the 2′-O-methylation at the ribose moiety (2′-O-me) and Ψ [47]. These modifications are installed by the methyltransferase fibrillarin (*FBL*) and pseudouridine synthase dyskerin (*DKC1*). In particular, modifications by FBL and DKC1 are guided by the complementarity of snoRNAs of the C/D-box (for 2′-O-me) and H/ACA-box (for Ψ) classes with target rRNA sequences [47]. Nucleotide modifications of rRNAs are important for both the synthesis and function of mature ribosomes [46]. Alteration in rRNA modification levels has been associated with several human diseases, including cancer. Changes in rRNA modification may impact ribosome synthesis and translation fidelity and increase cap-independent translation [48,49,50].

Despite their predicted housekeeping function, snoRNAs show specific expression in different lineages and developmental stages of hematopoiesis [51]. Furthermore, snoRNAs are highly expressed in acute myeloid leukemia (AML), and alteration in snoRNA levels was shown to play a relevant role in leukemogenesis [23,52]. In particular, the AML1-ETO fusion oncogenic protein, which characterized AML with the (8; 21) translocation, increased the expression of snoRNAs through the induction of amino-terminal enhancer of split (AES) protein expression [23] (Figure 2). Mechanistically, AES functions by interacting with the RNA helicase DDX21, which has an established role in regulating both rDNA transcription and rRNA processing and modification [53]. Interestingly, the effect of AES is specific for the C/D box snoRNAs. This results in high levels of rRNA 2′-O-me and increased protein translation in leukemic cells. Notably, AES is required for the oncogenic activity of AML1-ETO, and its depletion strongly impairs the capacity of AML1-ETO-expressing cells to produce leukemia in recipient mice [23]. The same study demonstrated that the deletion of single C/D box snoRNAs, *SNORD14D* or *SNORD35A*, in the MV4-11 leukemia cell line reduced cell survival and delayed leukemogenesis in xenotransplantation assays [23], indicating that the lack of single 2′-O-me modification in rRNA may already affect leukemic cells. The C/D box *SNORD42*, which guides the U116 2′-O-me in 18S rRNA, was recently found upregulated in AML [24]. Its depletion resulted in the reduction in 2′-O-me levels and a strong reduction in the translation of ribosomal proteins, which are required to maintain high levels of ribosomes and proliferation in AML cells [24]. Notably, this was the first demonstration that the lack of a single rRNA modification may affect ribosome function. However, it is still not known whether this is a general feature or a peculiarity of AML cells. Interestingly, the methyltransferase FBL is repressed by the tumor suppressor p53 (encoded by the *TP53* gene), which is one of the most frequently mutated genes in human cancer. In breast cancer, mutation in *TP53* resulted in increased *FBL* and rRNA 2′-O-me levels. Overexpression of FBL produced a decrease in translation fidelity and the specific IRES-dependent translation of pro-proliferative genes, such as IGF1R [49]. In AML, mutations of *TP53*, even if less frequent than other tumors, are associated with lowest survival rates [54]. Moreover, despite the presence of a wild-type *TP53* gene, AML patients present frequent inactivation of the p53 protein due to the upregulation of its negative regulators, such as MDM2 and MDM44 [54]. Therefore, as p53 protein regulates FBL activity and, eventually, 2′-O-me levels [49], this might result in altered levels of rRNA methylation and mRNA translation (Figure 2). In addition, it was recently demonstrated that FBL also interacts with EZH2, the catalytic component of the polycomb repressive complex 2 (PRC2) [55]. This interaction promotes the assembly of box C/D snoRNPs, increases 2′-O-me levels in rRNAs, and stimulates IRES-dependent translation. At present, this mechanism was only demonstrated in prostate cancer [55]. However, PRC2 plays an important role in the maintenance of HSCs and is frequently deregulated in hematological malignancies [56]. Thus, it will be interesting to validate this model in the hematopoietic compartment.

Mutations in the pseudouridine synthases gene *DKC1* are associated with dyskeratosis congenita (DC), a rare genetic disorder characterized by bone marrow failure and increased risk of developing hematological tumors. DC patients show reduction in Ψ levels of rRNAs [22]. DKC1 is also a component of the human telomerase ribonuclear proteins and is required for telomere stability. Most DC hallmarks are attributed to telomerase malfunction. However, CD34^+^ HSCs carrying a catalytic inactive DKC1 present defects in producing mature myeloid and erythroid cells in in vitro differentiation assays [22], thus indicating that the pseudouridine synthase activity of dyskerin is a requirement for hematopoietic differentiation. Moreover, different H/ACA snoRNAs are deregulated in hematological malignancies [52,57,58], suggesting that, similar to the snoRNAs of the box C/D class, they might play a role in the leukemogenesis process. In breast cancer, H/ACA snoRNAs are also able to regulate the activity of DDX21 by interacting with the PARP enzyme [59], which catalyzes the addition of poly(ADP-ribose) (PAR) polymers on target proteins [60]. PARP is involved in DNA repair and gene expression regulation and is a promising target for anticancer therapy [60,61]. The activation of PARP by snoRNAs ADPRylates the RNA helicase DDX21, which, in turn, stimulates rRNA production, ribosome levels, and translation in breast cancer cells [59]. Therefore, if conserved, this mechanism provides a positive loop in which H/ACA snoRNAs, similar to the effect of AES, stimulate the expression of C/D box snoRNAs via DDX21 and, eventually, rRNA methylation levels in leukemia (see above). Notably, the knockdown of *Ddx21* phenocopies the depletion of *Aes* by reducing growth capacity of leukemia cells [23].

An additional chemical modification with a role in hematopoietic development is the N^1^-methyladenosine (m^1^A) at position A_1309_ and A_1136_ in human and mouse 28S rRNA, respectively, which is required for the production of mature 60S ribosomal subunits [25]. This modification is installed by the methyltransferase nucleomethylin (*NML*; also known as *RRP8* in yeast). NML supports cell proliferation, and its knockdown in cell lines induces a p53-dependent cell cycle arrest and apoptosis [25]. *Nml* knockout mice were also produced [26]. NML depletion produced embryonic and early postnatal lethality in 80–90% of mice, indicating an important role for NML in embryonic development. The presence of *Nml* is critical for hematopoietic development. In particular, *Nml* depletion strongly impairs erythropoiesis in fetal liver cells [26]. Therein, it was suggested that NML is required for proper ribosome levels and that its depletion affects the production of proteins involved in erythropoiesis. Notably, it was shown that ribosome levels play a crucial role in erythroid lineage commitment from HSPCs [62], and in human diseases characterized by mutations in ribosomal proteins, such as Diamond-Blackfan anemia and 5q-syndrome, where the decrease in ribosome levels results in defective erythropoiesis [63]. Moreover, the defects in erythroid differentiation are often restored by disruption of *Tp53*. However, lethality and erythroid differentiation arrest in *Nml* knockout mice were not rescued by p53 suppression, indicating that p53-independent pathways must operate in *Nml*-depleted animals.

Other modifications in rRNAs, such as m^6^A and m^5^C, have been recently linked to ribosome function, stem cell differentiation, and cancer [27,28,29,64,65,66,67]. In particular, 18S and 28S rRNAs contain two m^6^A methylations at position A1832 and A4220, respectively. The lack of m^6^A modification on rRNAs does not affect pre-rRNA processing but is required for proper translation. The m^6^A_1832_ on 18S rRNA is installed by the METTL5 methyltransferase [27,28,66]. *Mettl5* depletion in mESCs has produced controversial results [27,28]. One study reported that METTL5 was required for pluripotency and global translational rate [27]. Moreover, *Mettl5* knock-out mice were sub-viable and presented developmental and behavioral defects [27]. A second study reported that METTL5 was dispensable for pluripotency and self-renewal of mESCs in vitro [28]. However, both studies found that METTL5 was required for correct differentiation. Furthermore, the lack of m^6^A in 18S rRNA was found to affect the translation of specific mRNAs [28]. Translation of *Fbxw7* (F-box and WD repeat domain-containing protein 7), a protein involved in the degradation of c-MYC, was found to be specifically regulated by 18S rRNA m^6^A modification and to play an important role in mESC differentiation [28]. Restoring *Fbxw7* expression compensated for the cell differentiation defect due to *Mettl5* depletion [28]. Notably, deregulation of *FBXW7* expression is connected to the development of different types of leukemia [68]. Therefore, it will be important to study the potential role of the m^6^A_1832_-FBXW7-c-MYC axis in hematological malignancies.

The m^6^A_4220_ in 28S rRNA is deposited by the ZCCHC4 methyltransferase [64]. Depletion of *ZCCHC4* in human cell lines (HeLa and HepG2) reduced the proliferation rate and affected the translation of different mRNAs [64]. Moreover, hepatocellular carcinoma tumor tissues presented high levels of *ZCCHC4* and m^6^A in 28S rRNA [64], indicating that ZCCHC4-mediated m^6^A methylation in 28S rRNA might play a general role in tumor growth and progression.

Another important modification for ribosome function with an established role in cancer is the m^5^C in rRNA. NSUN5 is the methyltransferase responsible for the installation of m^5^C at position C_3782_ and C_3438_ into human and mouse 28S rRNA, respectively [19]. *NSUN5* depletion in HeLa and HEK293 cell lines impaired cell proliferation and reduced global protein translation [19]. In this case, this was not due to defects in ribosome biogenesis. Consistent with its role in cell lines, *Nsun5* knockout mice are viable but present reduced body weight [19]. Despite its pro-proliferative effect, in human glioma tumors the *NSUN5* gene acts as an onco-suppressor and was found to be silenced by DNA methylation [20]. In particular, *NSUN5* silencing in glioma reduces global protein synthesis but results in the selective translation of specific mRNA to increase cell survival of stress conditions [20].

In conclusion, many different modifications of rRNAs have been linked to proper translational rate, cell differentiation, and cancer. However, their involvement in hematopoietic development and hematological malignancies has not yet been investigated. We predict that, in this context, this topic will be a prolific subject for further studies.

More recently, a novel high-throughput RNA sequencing methodology referred to as panoramic RNA display by overcoming RNA modification aborted sequencing (PANDORA-seq) has enabled the identification of ribosomal RNA-derived small RNA fragments (rsRNAs) [69]. Notably, rsRNAs present tissue- and cell-specific expression, thus indicating a possible role in cell identity. Moreover, the transfection of a 28S rRNA-derived rsRNA in mouse embryonic stem cells (mESCs) significantly reduced translation [69]. Thus, the impact of rRNA modifications on rsRNA biogenesis and their potential regulatory roles in normal and malignant hematopoiesis warrant further investigation.

## 4. Conclusions

Ribosome biogenesis and translation are tightly connected to cell growth and proliferation. In the hematopoietic compartment, these two processes are finely regulated during the differentiation of HSCs [12]. Moreover, altered protein synthesis levels can disrupt normal differentiation processes and promote the development and progression of hematopoietic malignancies [13,14]. In this scenario, two important players are rRNA and tRNA molecules. These two non-coding RNA species contain the highest number of chemical modifications and their dysregulation may cause an imbalance in protein synthesis and translation fidelity. Furthermore, modifications in these RNA molecules can also be important in linking the metabolic state of a cell to its translational output and adapting the translational outcome to particular stress conditions.

Though modifications in tRNA and rRNA molecules have been known for several years, our understanding of their importance in tRNA and rRNA function is still at an early stage. Furthermore, the biological function of the majority of RNA modifications remains to be characterized. We believe that in the coming years, we will expand our understanding of the importance of modifications on RNA and the mechanisms through which they act in cell differentiation and cancer. Moreover, the study of the enzymes involved in the nodulation of the epitranscriptome will enable us to find new candidates for drug targeting. In this context, hematopoietic development and hematological malignancies are paving the way.

## Figures and Tables

**Figure 1 ijms-22-04755-f001:**
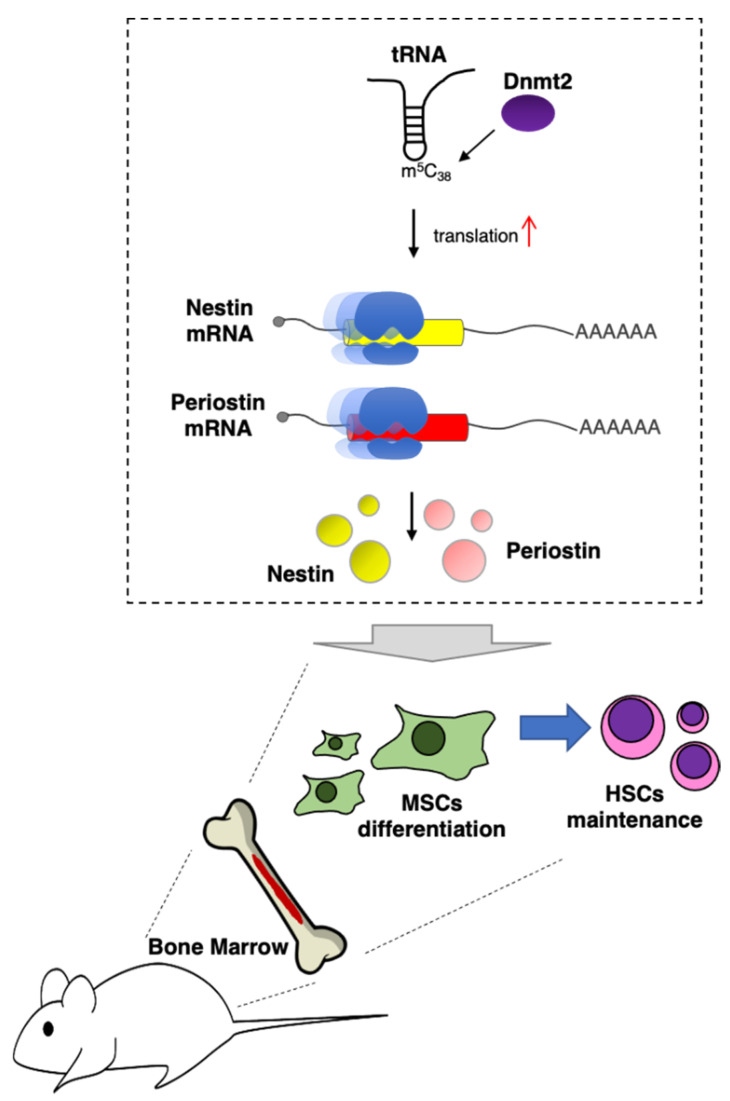
DNMT2 supports HSCs maintenance in vivo. DNMT2 installs m^5^C_38_ in tRNA molecules. This modification specifically increases translation of mRNA encoding for nestin and periostin proteins, which, in turn, are required for MSCs supporting activity of HSCs in bone marrow [23].

**Figure 2 ijms-22-04755-f002:**
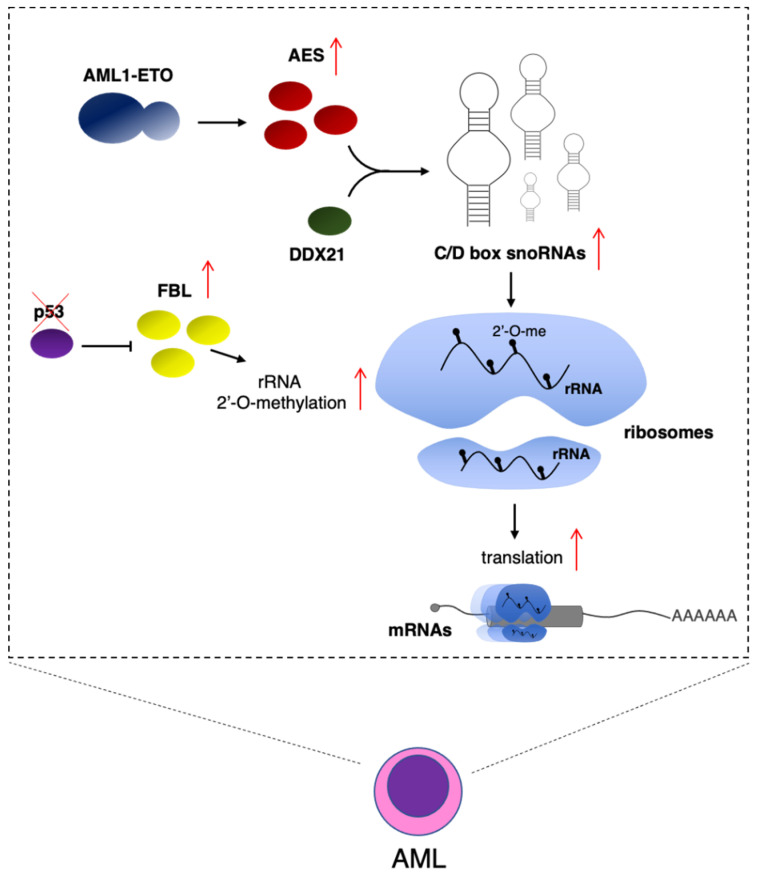
rRNA 2′-O-me levels regulate translation in AML cells. Expression of AML1-ETO in AML cells results in increased AES expression. AES interacts with the DDX21 RNA helicase to induce C/D box snoRNA levels. Mutation in *TP53*, which encodes for the p53 onco-suppressor, produces increased FBL levels. In both cases, this results in high levels of rRNA 2′-O-me and increased protein translation [39,43].

**Table 1 ijms-22-04755-t001:** Function of RNA modifications in tRNAs and rRNAs.

RNA Modification/RNA Species	Modifier (Type)	Genes or Pathway Affected	Cell Type/KO Phenotype
m^5^C/tRNAs	DNMT2 (writer)	nestin, periostin	MSCs, defect in differentiation [16]
m^5^C/tRNAs	TET2 (eraser)	SOCS3	HSCs, defect in myelopoiesis [17,18]
m^5^C/28S rRNA	NSUN5 (writer)	translation	Human cell lines, proliferation defects; glioma, increase cell surviving [19,20]
Ψ/tRNAs	PUS7 (writer)	tRNA fragments	HSCs, defect in differentiation [21]
Ψ/rRNAs	DKC1 (writer)	translation	HSCs, defect in differentiation [22]
2′-O-me/rRNAs	FBL (writer)	ribosomal protein	AML cell line, reduced cell survival [23,24]
m^1^A/28S rRNA	NML	60S ribosomal subunits	AML cell line, cell cycle arrest and apoptosis; fetal liver cells, defect in differentiation [25,26]
m^6^A/18S rRNA	METTL5 (writer)	FBXW7	mESCs, differentiation defects [27,28]
m^6^A/28S rRNA	ZCCHC4 (writer)	translation	HCC cell line, proliferation defects [29]

HCC, hepatocellular carcinoma cell; HSCs, hematopoietic stem cells; MSCs, mesenchymal stromal cells; mESCs, mouse embryonic stem cells.

## Data Availability

Not applicable.

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
