# Peer review of "Regulation of Ribosome Function by RNA Modifications in Hematopoietic Development and Leukemia: It Is Not Only a Matter of m6A"

_ijms, 2021, doi:10.3390/ijms22094755_

Round 1

Reviewer 1 Report

Very well written review on RNA modifications, that addresses several novel aspects to better understand the (potential) role of certain RNAs in the development of cancer. The authors comment at few sites throughout the manuscript on the potential in AML and hematopoiesis. The author's discussion and conclusion / concluding hypothesis are well worded.

Figure 1 and Figure 2 are a little bit too simplistic and a more detailed design would be helpful. Also, if the figure legends would include the references would be helpful to the reader.

The authors may want to consider citing Mohammad Burhan Uddin's work in Theranostics 2020 Vol 10 Issue 7, as this article discusses several RNA modifications and their role in cancer / AML.

Author Response

Figure 1 and Figure 2 are a little bit too simplistic and a more detailed design would be helpful. Also, if the figure legends would include the references would be helpful to the reader.

We have modified the figures and included references in the figure legends.

The authors may want to consider citing Mohammad Burhan Uddin's work in Theranostics 2020 Vol 10 Issue 7, as this article discusses several RNA modifications and their role in cancer / AML.

We added the reference as suggested by the referee.

Reviewer 2 Report

In this review, Dr Fazi and Fatica summarize the RNA modifications in tRNA and rRNA molecules that affect regulation of normal and malignant hematopoietic development. The authors cite relevant papers from the fied and report examples of tRNA and rRNA modifications, including alterations of DNMT2, TET2, PUS7, FBL, DKC1 and METTL5.

The manuscript is well written, but I have few comments to improve its quality and clarity.

  • Page 6, lines 240-243: “Moreover, despite the presence of a wild-type TP53 gene, AML patients present frequent inactivation of p53 protein due to the upregulation of its negative regulators, such as Mdm2 and Mdm4 [44].Therein, this might result in altered levels of rRNA methylation”. The reasons of this statement are not completely clear and supported by references. Please explain mechanisms and cite appropriate literature.
  • A Table summarizing and describing the type of RNA modification, the genes affected, the type of cells and references would be extremely helpful. It is hard to follow the list of all examples just in the text.
  • Please revise nomenclatures for human and mouse genes and proteins. These are not correct throughout the manuscript.
  • A recent review describes similar modifications of RNA occurring in cancer and a citation it might be helpful for the reader: “RNA modifications regulating cell fate in cancer Sylvain Delaunay & Michaela Frye, Nature Cell Biology 2019.

Author Response

  • Page 6, lines 240-243: “Moreover, despite the presence of a wild-type TP53 gene, AML patients present frequent inactivation of p53 protein due to the upregulation of its negative regulators, such as Mdm2 and Mdm4 [44].Therein, this might result in altered levels of rRNA methylation”. The reasons of this statement are not completely clear and supported by references. Please explain mechanisms and cite appropriate literature.

 We thanks the reviewer for the suggestion. We added a sentence explaining the mechanism and reference.

  • A Table summarizing and describing the type of RNA modification, the genes affected, the type of cells and references would be extremely helpful. It is hard to follow the list of all examples just in the text.

We thanks the reviewer for the suggestion. We added a Table (table 1) that summarises type of RNA modifications,  targets and effects. 

  • Please revise nomenclatures for human and mouse genes and proteins. These are not correct throughout the manuscript.

We have revised nomenclature for gene and proteins.

  • A recent review describes similar modifications of RNA occurring in cancer and a citation it might be helpful for the reader: “RNA modifications regulating cell fate in cancer Sylvain Delaunay & Michaela Frye, Nature Cell Biology 2019.

We added the suggested reference.

Round 2

Reviewer 2 Report

The authors addressed my comments and I do not have further questions.